# Muscle-specific AXIN1 and AXIN2 double knockout does not alter AMPK/mTORC1 signalling or glucose metabolism

Kaspar W. Persson[1], Roberto Meneses-Valdés[1], Nicoline R. Andersen[1], Frederik S. Pedersen[1], Samantha Gallero[1], Sofie A. Hesselager[1], Carlos Henriquez-Olguin[1,2] and Thomas E. Jensen[1] (iD)

[1]*August Krogh Section for Human and Molecular Physiology, Department of Nutrition, Exercise and Sports, Faculty of Science, University of Copenhagen, Copenhagen, Denmark*
[2]*Exercise Science Laboratory, Faculty of Medicine, Universidad Finis Terrae, Santiago, Chile*

Handling Editors: Paul Greenhaff & Bettina Mittendorfer

The peer review history is available in the Supporting Information section of this article (https://doi.org/10.1113/JP288854#support-information-section).

**The Journal of Physiology**

**Abstract figure legend** This study investigated the role of AXIN1 and AXIN2 in regulating skeletal muscle AMP-activated protein kinase (AMPK) and mechanistic target of rapamycin complex 1 (mTORC1) signalling and glucose uptake. Although AXIN1 has been proposed as a key scaffold linking energy sensing to catabolic and anabolic signalling pathways in cell systems – and implicated in GLUT4 translocation – muscle-specific AXIN1/2 double knockout mice displayed normal AMPK and mTORC1 signalling in response to 5-aminoimidazole-4-carboxamide ribonucleotide (AICAR), insulin and contraction, as well as normal glucose uptake after AICAR and insulin. These findings suggest that AXIN proteins are dispensable for acute regulation of AMPK, mTORC1 and glucose metabolism in skeletal muscle.

**Kaspar W. Persson** obtained his BSc in sports science and MSc in human nutrition from the University of Copenhagen. He is completing his PhD in molecular physiology in 2025 under the supervision of Professor Thomas E. Jensen, focusing on imaging-based investigations of GLUT4 dynamics in adult skeletal muscle. His research centres on the regulation of muscle growth and glucose metabolism in health and disease, with a particular focus on subcellular signalling mechanisms and protein localization.

**Abstract**   AMP-activated protein kinase (AMPK) and mechanistic target of rapamycin complex 1 (mTORC1) are crucial kinase signalling hubs that regulate the balance between catabolism and anabolism in skeletal muscle. The scaffold protein AXIN1 has been proposed to regulate the switch between these pathways and be required for GLUT4 translocation in skeletal muscle and adipocyte cell lines. Muscle-specific AXIN1 knockout (KO) mice exhibit no discernable phenotype, possibly due to compensation by AXIN2 upon AXIN1 loss. Thus we generated and characterized muscle-specific inducible AXIN1 and AXIN2 double knockout (dKO) mice. Surprisingly AXIN1/2 dKO mice displayed normal AMPK and mTORC1 signalling and glucose uptake in response to 5-aminoimidazole-4-carboxamide ribonucleotide (AICAR), insulin and *in situ* muscle contraction. These findings suggest that AXIN proteins are not essential for the regulation of AMPK and mTORC1 signalling or glucose uptake in skeletal muscle. This study challenges the previously indicated critical roles of AXIN1 in exercise-stimulated AMPK activation and GLUT4-mediated glucose uptake in skeletal muscle.

(Received 6 March 2025; accepted after revision 11 June 2025; first published online 29 June 2025)

**Corresponding author** Thomas E. Jensen: August Krogh Section for Human and Molecular Physiology, Department of Nutrition, Exercise and Sports, Faculty of Science, University of Copenhagen, Copenhagen, Denmark.    Email: TEJensen@NEXS.ku.dk

**Key points**

- Phenotyping of tamoxifen-inducible muscle-specific AXIN1/2 double knockout (dKO) mice.
- We find no evidence for AXIN-dependent AMPK or mTORC1 regulation in skeletal muscle by insulin, AMPK activation or contraction.
- Glucose uptake regulation by insulin and AMPK activation is normal in AXIN1/2 dKO mice.

# Introduction

AMP-activated protein kinase (AMPK) and mechanistic target of rapamycin complex 1 (mTORC1) are critical kinase signalling hubs that regulate the cellular transition between catabolism and anabolism (González et al., 2020). In skeletal muscle fibres these two pathways have been linked in many contexts to muscle homeostasis and function, responding dynamically to changes in energy availability, nutrient status and hormones in a reciprocal, mutually exclusive manner, but with context-dependent regulation that allows for simultaneous activity under certain physiological conditions (Knudsen et al., 2020; Li et al., 2025). Understanding the molecular mechanisms by which AMPK and mTORC1 signalling are regulated and co-ordinated may reveal new potential drug targets for use in skeletal muscle fibres or other cell types.

In 2014 the scaffold protein Axis inhibitor protein 1 (AXIN1) was suggested to be part of a late endosomal/lysosomal protein complex regulating the reciprocal switch between pro-catabolic AMPK and pro-anabolic mTORC1 signalling in multiple cell types, including skeletal muscle (Zhang et al., 2014). In brief the model established in the original and subsequent papers postulated that v-ATPase sensing of low glucose recruited AXIN1 to the lysosomal surface, whereby the AXIN1-bound AMPK-activating kinase liver kinase B1 (LKB1) could phosphorylate and activate lysosomal AMPK by an AMP/ADP-independent mechanism (Li et al., 2019; Zhang et al., 2013, 2017, 2022). Simultaneously AXIN1 inhibited the guanine nucleotide exchange factor activity of Ragulator against Rag GTPases to cause lysosomal dissociation and inactivation of mTORC1 (Zhang et al., 2014). In skeletal muscle knockout of the AXIN1-binding Ragulator subunit LAMTOR1 prevented treadmill-exercise-stimulated AMPK activation in gastrocnemius muscle (Zhang et al., 2014), suggesting that this mechanism remarkably accounted for most, if not all, of exercise/contraction-stimulated AMPK activation in skeletal muscle. In addition to its proposed role in AMPK and mTORC1 signalling, AXIN1 was later suggested to regulate GLUT4 translocation by promoting GTP loading of Rac1 by *in vitro* electrical stimulation in C2C12 mouse myotubes (Yue et al., 2020), a well-evidenced requirement for skeletal muscle insulin and exercise-stimulated GLUT4 translocation (Henríquez-Olguin et al., 2019; Khayat et al., 2000; Sylow et al., 2016; Sylow, Jensen, Kleinert, Højlund et al., 2013; Sylow, Jensen, Kleinert, Mouatt et al., 2013; Ueda et al., 2010). Furthermore AXIN1 was shown to be required for insulin-stimulated KIF3A-dependent GLUT4 trans-

location in cultured 3T3-L1 adipocytes (Guo et al., 2012). Thus AXIN1 has been proposed to impact skeletal muscle AMPK and mTORC1 signalling and glucose metabolism in multiple ways.

To directly investigate if AXIN1 is critical to skeletal muscle function we previously comprehensively characterized muscle-specific tamoxifen-inducible AXIN1 knockout (KO) mice (Li et al., 2021). Despite the absence of AXIN1 these mice showed completely normal regulation of energy substrate metabolism, skeletal muscle AMPK and mTORC1 signalling and muscle glucose uptake. Meanwhile AXIN1 has a mammalian homologue AXIN2 ($\sim$44% sequence homology; Chia & Costantini (2005)), which was reported to be functionally redundant in the regulation of AMPK (Zong et al., 2019). Because AXIN2 is upregulated in skeletal muscle during myogenesis (Huraskin et al., 2016), a formal possibility was that compensation by AXIN2 explained the normal cell signalling and metabolic regulation in muscle-specific AXIN1 KO mice.

To formally address this possibility we presently characterized skeletal muscle-specific inducible AXIN1 and AXIN2 double KO (dKO) mice. We report that the dKO mice, like the muscle-specific AXIN1 single KO, are indistinguishable from the wild-type (WT) littermates in terms of cell signalling and glucose uptake regulation, indicating that AXIN proteins are dispensable for skeletal muscle AMPK and mTORC1 signalling and metabolic regulation.

## Methods

### Ethical approval

All experiments and the breeding protocol were approved by the Danish Animal Experiments Inspectorate (license: 2017-15-0201-0 1311) and conducted in accordance with the European Convention for the Protection of Vertebrate Animals used for Experimental and Other Scientific Purposes.

### Animals

Male and female muscle-specific AXIN1/2 KO (imKO) mice were generated using a tamoxifen-inducible Cre-recombinase driven by the human $\alpha$1-skeletal actin promoter (HSA-MCM$\pm$) (McCarthy et al., 2012) on a C57Bl/6NRj background purchased from Janvier Labs. Homozygous floxed littermates served as WT controls. At 10–14 weeks of age, Cre recombinase was activated by three consecutive intraperitoneal injections of 70 mg/kg body weight tamoxifen (cat. no. T5648, Sigma Aldrich) dissolved in sunflower seed oil (cat. no. S5007, Sigma Aldrich) every 48 h.

Mice were group-housed under standard conditions at 22°C–24°C with a 12-h light/dark cycle and *ad libitum* access to water and chow diet. *In situ* contractions and *ex vivo* 5-aminoimidazole-4-carboxamide ribonucleotide (AICAR) muscle incubations were conducted 4–8 weeks after the first tamoxifen injection, whereas insulin-stimulated muscle incubations were performed 12 weeks postinjection. Mice were used as follows: AICAR incubations included 18 WT (10 females, 8 males) and 20 imKO (10 females, 10 males); insulin incubations included 4 WT and 4 imKO (all females); and *in situ* contractions included 16 WT (9 females, 7 males) and 14 imKO (7 females, 7 males). All procedures conformed to the principles of *The Journal of Physiology* for animal research ethics and welfare.

### Genotyping

Genotyping was performed as previously described (Li et al., 2021). The procedure was identical to that described in our previous AXIN1 study. The primer sequences were as follows:

AXIN1 wt (sense: GCATTGAGAATGCAACAAACAA GACT/anti-sense: TCCATTCTCCTGCCTTAGCTTCC), AXIN1 flox (sense: ATAGCATTGAGAATGCAACAAAC AAG/anti-sense: TTGCATGCCTGCAGGTCGAAGA), AXIN2 wt (sense: TTGAGAGCTAGGGCTTCTGGCTAG G/anti-sense: AACAAAACATGAGATCAAAGGGTTCC), AXIN2 flox (sense: AGGGTTGATTGAGAGCTAGGGCT TCTG/anti-sense: GAAGTTCCTATTCCGAAGTTCCT ATTCTC), Cre (sense: ACGGACAGAAGCATTTTCCAG GT/anti-sense: CGGTCGATGCAACGAGTGATG).

### *Ex vivo* muscle incubation and 2-deoxyglucose uptake

*Ex vivo* muscle incubation experiments were performed as previously described (Jensen et al., 2007; Li et al., 2021). In brief soleus and extensor digitorum longus (EDL) muscles were excised from mice anesthetized by intraperitoneal injection of sodium pentobarbital (6 mg) and lidocaine (0.2 mg) per 100 g body weight. Adequate anaesthesia was confirmed by the absence of tail pinch and pedal withdrawal reflexes. After muscle excision animals were killed by cervical dislocation. Excised muscles were maintained at resting length in incubation chambers (Multi Myograph System, Danish Myo-Technology) in oxygenated (95% $O_2$, 5% $CO_2$) Krebs-Ringer-Henseleit (KRH) buffer with 2 mM pyruvate and 8 mM mannitol for 30 min prior to glucose uptake measurements. To assess AICAR- and insulin-induced 2-deoxyglucose (2-DG) uptake muscles were stimulated with or without 4 mM AICAR for 30 min or 60 nM insulin for 10 min. Subsequently 2-DG (1 mM) and [$^3$H]-2-DG (0.125 $\mu$Ci/ml) were added for 10 min to measure glucose uptake

with [14C]-mannitol (0.14 µCi/ml) included to correct for extracellular space. 2-DG uptake was terminated by washing muscles in ice-cold KRH buffer before snap-freezing in liquid nitrogen. Muscle lysates were analysed for radioactivity by scintillation counting.

### *In situ* muscle contraction

Unilateral *in situ* contractions were performed on anesthetized mice (6 mg pentobarbital and 0.2 mg lidocaine per 100 g body weight) placed on a heating pad. Adequate anaesthesia was confirmed by the absence of tail pinch and pedal withdrawal reflexes prior to the procedure. The quadriceps muscle was electrically stimulated via acupuncture needles (0.2 mm; TAI-CHI; B.C. Medical, Nykøbing SJ, Denmark) inserted into the proximal and distal regions of the muscle. The contra-lateral resting control leg had needles identically inserted but remained unstimulated. The contraction protocol consisted of nine sets of electrical stimulations at 10 V with a pulse duration of 0.1 ms at 100 Hz lasting for 3 s and starting every 10 s. Each set lasted 1 min with 30 s of rest between sets, as previously described (Knudsen et al., 2020). Following the contraction protocol the skin was rapidly removed from the muscles, and the muscles were freeze-clamped *in situ* using liquid nitrogen-cooled tongs. The frozen part was then dissected and snap-frozen in liquid nitrogen. Mice were killed by cervical dislocation under anaesthesia following the procedures outlined for *ex vivo* incubations.

### Tissue processing

Soleus and EDL were trimmed to remove tendons and weighed prior to homogenization for 2-DG uptake assays and western blot analysis. Quadriceps muscles were crushed in a disposable plastic tray submerged in liquid nitrogen using liquid nitrogen-cooled tongs, and 10–15 mg muscle powder of each muscle was collected for homogenization. Muscle homogenization, protein extraction and determination were performed as previously described (Li et al., 2021), and resulting lysates were adjusted to equal amounts of protein in MilliQ water and mixed with Laemmlii sample buffer (62.5 mm Tris (pH 6.8), 2% SDS, 10% glycerol, 0.1 m DTT, 0.01% bromophenol blue) for western blot analysis. One-third of the 300 µl soleus and EDL lysates were used for 2-DG uptake measurements by liquid scintillation counting.

### Western blotting

Proteins were separated by SDS-PAGE on 7%–15% gels and transferred to PVDF membranes in a semi-dry blotting system (Bio-Rad). Membranes were blocked at room temperature in TBS-T with 3% BSA for 15 min before overnight incubation in primary antibodies at 4°C. In the following day the membranes were washed in TBS-T and incubated in HRP-conjugated secondary antibody (cat no. 111-035-045, Jackson ImmunoResearch, 1:3000 in 3% BSA) for 45 min followed by additional washes in TBS-T. Target proteins were visualized using enhanced chemiluminescence (Immobilon Forte Western HRP Substrate, Millipore) in a Bio-Rad Chemidoc MP imaging system. Band intensities were quantified using Image Lab software (version 6.1) with the volume tool and normalized to the gel average. To indicate equal protein loading Coomassie staining was performed after chemiluminescent image acquisition. Primary antibodies were used at 1:1000 unless otherwise specified and included AXIN1 (#2087, Cell Signalling Technology (CST)), phospho(p)-p70S6K1 Thr389 (#9205, CST), p-rpS6 Ser235/236 (#2211, CST), p-ribosomal protein S6 Ser240/244 (#2215, CST) and p-Raptor Ser792 (1:500) (#2083, CST) in 3% BSA and p-AKT Thr308 (#9275, CST) p-ACC Ser212 (#03-303, Millipore) and p-AMPK Thr172 (#2535, CST) in 3% skimmed milk.

### Statistics

All data are presented as individual, paired data points, with males shown in blue and females in pink, alongside mean values for both sexes. Student's *t* test or two-way ANOVA was performed to assess the effects of genotype and treatment, with analyses conducted separately for each sex and for the combined dataset. When a significant main effect or interaction was detected ($P$-value $<0.05$), Sidak's *post hoc* test was applied for multiple comparisons. Statistical analyses were performed using GraphPad Prism 10.4.1 (GraphPad, CA, USA).

## Results

### Validation of the muscle-specific AXIN1/2 dKO model

AXIN1 and AXIN2 homogeneous double-floxed mice, around 10–14 weeks of age with or without tamoxifen-inducible muscle-specific Cre recombinase (McCarthy et al., 2012), were injected with tamoxifen (70 mg/kg) thrice over a week and investigated minimum 4 weeks later (Fig. 1*A*). At this time point AXIN1 and AXIN2 showed ~40% floxed allele excision in quadriceps muscle (Fig. 1*B* and *C*), as expected from skeletal muscle where muscle fibre nuclei constitute approximately 50% of total nuclei (Madsen et al., 2018). Although the excision effect size was slightly lower than the ~50% previously observed in muscle-specific AXIN1 KO mice (Li et al., 2021) and other muscle-specific KO models (Knudsen et al., 2022; Madsen et al., 2018), it was

sufficient for near-complete depletion of AXIN1 protein levels (Figs 2*C*; 3*B*, 4*C*) assessed by western blot analysis, consistent with our previously muscle-specific AXIN1 KO mouse study (Li et al., 2021). Despite comprehensive testing of commercially available AXIN2 antibodies none demonstrated sufficient specificity for reliable protein detection (Li et al., 2021). Still the level of AXIN1 and AXIN2 gene excision suggested efficient KO of AXIN1 on par with our previous study and clear excision of AXIN2, suggesting successful generation of a muscle-specific AXIN1/2 dKO mouse model.

### AXIN1/2 dKO does not disrupt AMPK activation and glucose uptake upon AICAR stimulation

We proceeded to investigate the effect of AXIN1/2 dKO on skeletal muscle cell signalling and glucose metabolism in male and female cohorts of mice. First *ex vivo*-incubated fast-twitch EDL and slow-twitch soleus muscles were stimulated with the classical AMP-mimetic activator AICAR (Fig. 2*A*). Overall AXIN1/2 KO mice showed clearly reduced AXIN1 protein expression in EDL, whereas soleus showed higher residual AXIN1 expression (Fig. 2*B*), consistent with our previous publication in muscle-specific AXIN1 KO mice (Li et al., 2021) and confirming successful tamoxifen-induced gene excision for the specific experiment. Despite the loss of AXIN proteins AXIN1/2 dKO mice exhibited normal *ex vivo*-unstimulated and AICAR-stimulated 2-DG uptake (Fig. 2*B*) and AMPK phosphorylation and downstream signalling (Fig. 2*D–F*), as well as normal AICAR-mediated suppression of mTORC1-dependent p70S6K signalling (Fig. 2*G*). Some sex-specific genotype effects were observed but were not reproducible between independent AICAR-incubation experiments and not

observed for the unstimulated condition in the subsequent insulin-incubation experiment (Fig. 3). Based on these observations AMPK activation by AICAR and its suppression of the mTORC1-p70S6K pathway do not require AXIN proteins in mouse skeletal muscle.

### Normal insulin-stimulated glucose uptake and mTORC1 signalling in AXIN1/2 KO mice

Next we measured insulin-stimulated glucose uptake and activation of the Akt-mTORC1-S6 signalling axis in incubated soleus and EDL skeletal muscle from female mice (Fig. 3*A*). As expected AXIN1 protein levels were reduced in KO mice, confirming tamoxifen-induced gene excision (Fig. 3*B*). No genotype differences were observed for basal or insulin-stimulated 2-DG uptake or Akt Thr308 phosphorylation nor for p70S6K Thr389 phosphorylation as a readout of mTORC1 activity (Fig. 3*C–E*). Ribosomal S6 phosphorylation on Ser235/236 also responded normally in AXIN1/2 dKO mice (Fig. 3*F*), whereas insulin-stimulated S6 Ser240/244 phosphorylation showed a small but significant increase in EDL and decrease in soleus, respectively (Fig. 3*G*). Thus *ex vivo* basal and insulin-stimulated glucose uptake, Akt and mTORC1 signalling in skeletal muscle function normally in the absence of AXIN proteins.

### AXIN1/2 KO does not affect contraction-induced AMPK and mTORC1 signalling

Finally we performed acute *in situ* contraction of quadriceps femoris by unilateral electrical stimulation (Fig. 4*A*). Muscles were freeze-clamped *in situ* and excised immediately after the protocol to preserve contraction-induced signalling. AXIN1 protein

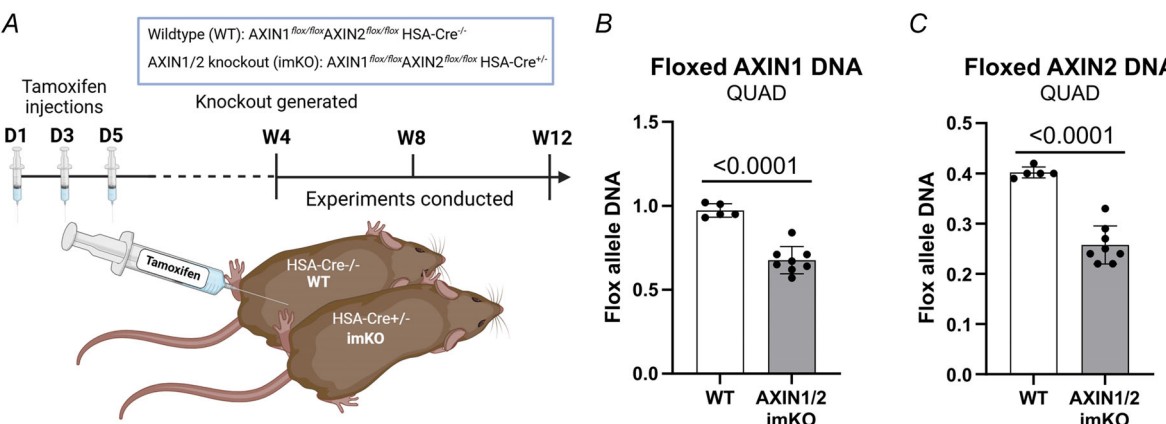

**Figure 1. Verification of skeletal muscle AXIN1/2 double knockout**
*A*, schematic overview of the tamoxifen induction protocol. *B, C*, quantification of the *Axin1* and *Axin2* flox allele excision in quadriceps muscle. Data are presented as means ± standard deviation with individual data points. Statistical analysis was performed using Student's *t* test. *n* = 5–8. [Colour figure can be viewed at wileyonlinelibrary.com]

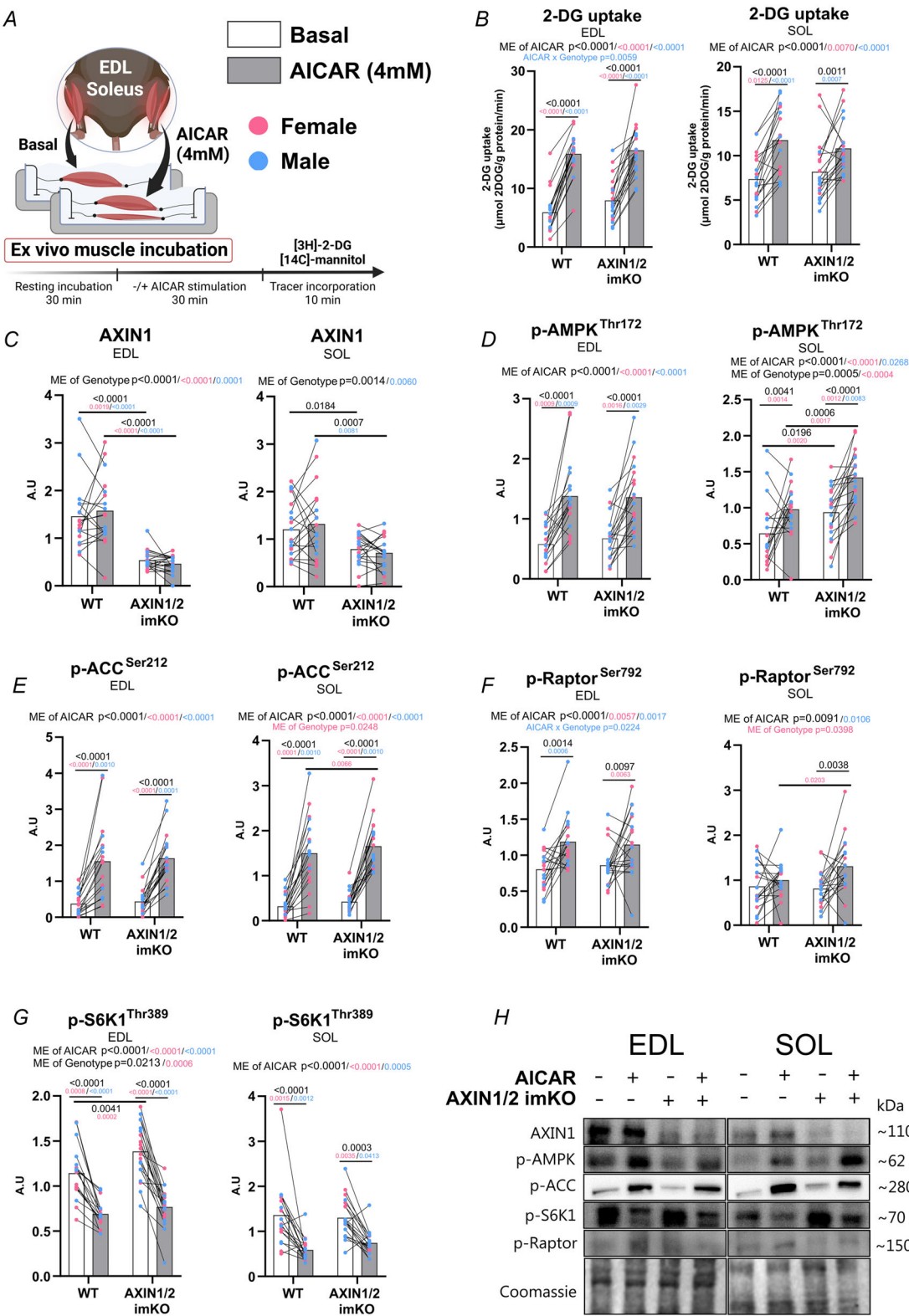

**Figure 2. AXIN1 and AXIN2 knockout does not affect *ex vivo* 5-aminoimidazole-4-carboxamide ribonucleotide (AICAR)-stimulated 2-deoxyglucose (2-DG) uptake and downstream AMPK and mechanistic target of rapamycin complex 1 (mTORC1) signalling**

*A*, experimental overview of *ex vivo* AICAR-stimulated 2-DG uptake in incubated extensor digitorum longus (EDL) and soleus (SOL) muscle. *B*, *ex vivo* AICAR (4 mM)-stimulated 2-DG uptake. *C–G*, levels of AMPK- and mTORC1-related phosphorylated protein after AICAR stimulation. *H*, representative blots of EDL and SOL muscle.

Data are presented as means with sex-colour-coded, paired individual values (pink: female; blue: male). Sex-specific (pink/blue) and pooled data (black) P-values were derived from two-way ANOVA with Sidak's multiple comparisons test in case of a significant ANOVA main effect (ME). $n$ = 18–20 (males: 8–10; females: 10). [Colour figure can be viewed at wileyonlinelibrary.com]

expression was lowered by >80% in AXIN1/2 dKO quadriceps compared to WT (Fig. 4*B*). Basal and electrically stimulated AMPK signalling remained normal in AXIN1/2 dKO mouse quadriceps (Fig. 4*C–E*). mTORC1-dependent p70S6K signalling did not respond to contractions in this experiment but again showed no genotype differences (Fig. 4*F*). No sex-specific differences were observed. These data show that skeletal muscle AMPK activation by contractile activity *in situ* does not require AXIN proteins.

## Discussion

Lysosomal AXIN-AMPK signalling has been suggested to account for most exercise-stimulated AMPK activation in skeletal muscle and to suppress mTORC1 signalling to co-ordinate the anabolic to catabolic transition (Zhang et al., 2014). AXIN2 shows genetic redundancy with AXIN1 in many instances, and AXIN2 has been reported to compensate for AXIN1 in lysosomal AMPK signalling in HEK293T cells (Zong et al., 2019). Furthermore AXIN has been linked to Rac1 and KIF3A-dependent GLUT4 translocation to the plasma membrane in cell culture (Guo et al., 2012; Yue et al., 2020). Therefore we currently investigated muscle-specific AXIN1/2 dKO mice as a follow-up to our previous muscle-specific AXIN1 KO study (Li et al., 2021). Similar to the AXIN1 KO mice, dKO mice display overall normal AMPK, Akt and mTORC1 signalling and normal glucose uptake stimulation by AMPK or insulin. Thus we find no evidence in mouse skeletal muscle supporting the existence of the lysosomal AXIN-AMPK pathway nor supporting the involvement of AXIN proteins in GLUT4-dependent glucose uptake regulation.

AXIN proteins were proposed to be involved in an energy stress-level-dependent, compartment-specific activation of AMPK (Zong et al., 2019). According to the 'hierarchical model', glucose starvation in the absence of AMP increases lowers glycolytic fructose 1,6-bisphosphate (FBP) binding to aldolase, which triggers formation of a lysosomal Axin-dependent complex promoting AMPK activation. Moderate energy stress elevates AMP and activates AXIN-dependent cytosolic and lysosomal AMPK, and severe energy stress activates cytosolic and mitochondrial AMPK independently of AXIN (Zong et al., 2019). Because the AXIN-AMPK pathway dominates at low-to-moderate energy stress, it is formally possible that the AICAR dose used here was too high to reveal the AXIN-dependent

mechanism. Still no genotype difference was detectable in response to *in situ* contraction nor were there any signs of abnormal resting or stimulated mTORC1 signalling. Together these findings argue against a significant role for AXIN-dependent AMPK regulation in contraction-induced AMPK activation or suppression of mTORC1 signalling in skeletal muscle. AXIN1 has also been linked to GLUT4-dependent glucose uptake in two studies. In one study acute AMPK activation by electrical stimulation in C2C12 myotubes or by intra-muscular AICAR injection *in vivo* upregulated AXIN expression in the myotubes and mouse gastrocnemius after 60 min, respectively (Yue et al., 2020). Furthermore small interfering RNA (siRNA) knockdown of AXIN1 in C2C12 myotubes lowered AMPK phosphorylation, whereas knockdown of AMPK or AXIN1 prevented Rac1 GTP-loading, GLUT4 translocation and glucose uptake by electrical stimulation. This suggested the existence of an AMPK-AXIN-Rac1-GLUT4 pathway. In a second study in 3T3L1 adipocytes insulin stimulated the interaction of AXIN1 with the ADP-ribosylase Tankyrase 2 and the microtubule-based kinesin motor protein KIF3A, a complex required for insulin-stimulated GLUT4 translocation (Guo et al., 2012). In our present study AMPK and insulin-stimulated glucose uptake was completely normal in mouse muscles lacking AXIN1 and AXIN2. In addition we saw no evidence of acute changes in AXIN1 expression in our present 40 min AICAR incubation experiments, consistent with our previous work (Li et al., 2021). Also our mouse muscle data in other publications generally favour a model where AMPK and Rac1 signal in parallel rather than in series to promote GLUT4-dependent skeletal muscle glucose uptake, as extensively discussed in our AXIN1 KO publication (Li et al., 2021). In summary the normal glucose uptake regulation in muscle-specific AXIN1/2 dKO mice does not support the previously reported requirement of AXIN1 for GLUT4 translocation and glucose uptake stimulation nor did we observe any signs of AMPK-stimulated AXIN1 expression.

Compared to our previous characterization of AXIN1 KO mice we performed a less-elaborate characterization of AXIN1/2 dKO mice for ethical and financial reasons. Also we focused on AMPK, mTORC1 and glucose uptake regulation, and did not examine the multiple other signalling pathways linked to AXIN proteins, including WNT/$\beta$-catenin, Hippo, TGF$\beta$, MAPK, NRF2 and cGAS/STING signalling pathways (Qiu et al., 2024). A deeper phenotypic or omics-based investigation may have revealed some genotype-dependent effects in other intra-

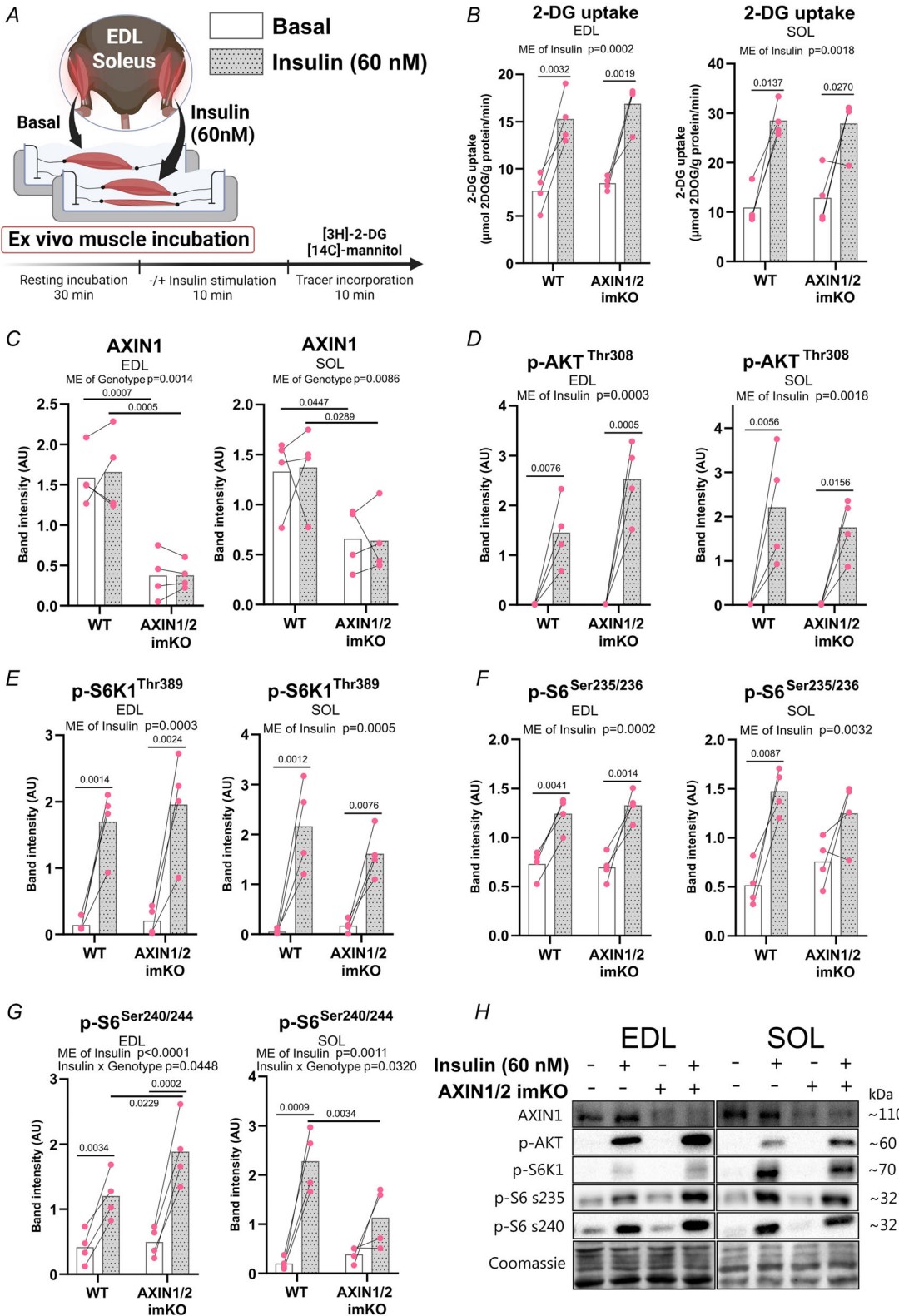

**Figure 3. AXIN1 and AXIN2 knockout does not affect *ex vivo* insulin-stimulated 2-deoxyglucose (2-DG) uptake and downstream AMPK and mechanistic target of rapamycin complex 1 (mTORC1) signalling**
*A*, experimental overview of *ex vivo* insulin-stimulated 2-DG uptake in incubated extensor digitorum longus (EDL) and soleus (SOL) muscle. *B*, *ex vivo* insulin (60 nM)-stimulated 2-DG uptake. *C–G*, levels of mTORC1-related phosphorylated protein after insulin stimulation. *H*, representative blots of EDL and SOL muscle. Data are presented

as means with paired individual values and analysed using two-way ANOVA with Sidak's multiple comparisons test in case of ANOVA main effect (ME). *n* = 4 (females). [Colour figure can be viewed at wileyonlinelibrary.com]

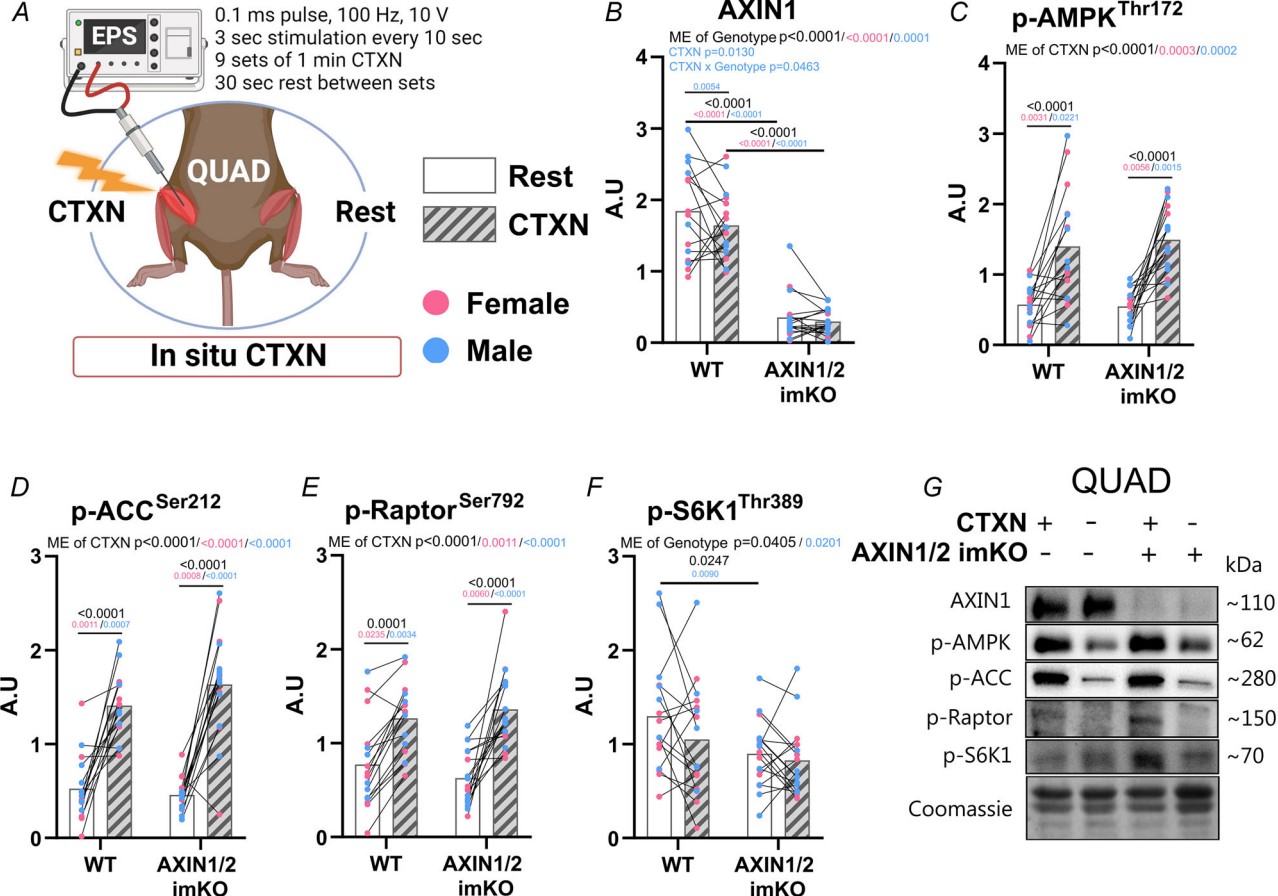

**Figure 4. Contraction-induced AMPK and mechanistic target of rapamycin complex 1 (mTORC1) signalling is not affected by AXIN1 and AXIN2 muscle knockout**

*A*, experimental overview of unilateral, percutaneous *in situ* contraction (CTXN) in quadriceps (QUAD) muscle. *B–F*, levels of AMPK and mTORC1-related phosphorylated protein acutely after *in situ* CTXN. *G*, representative blots of QUAD muscle. Data are presented as means with sex colour-coded, paired individual values (pink: female; blue: male). Sex-specific (pink/blue) and pooled data (black) *P*-values were derived from two-way ANOVA with Sidak's multiple comparisons test in case of ANOVA main effect (ME). *n* = 16–18 (males: 7–10; females: 7–9). [Colour figure can be viewed at wileyonlinelibrary.com]

cellular pathways. This may be perceived as a limitation of the current study.

In conclusion AXIN1 and AXIN2 are dispensable for mouse skeletal muscle AICAR, insulin and contraction-stimulated AMPK activation, mTORC1 inhibition by AICAR and stimulation by insulin and glucose uptake stimulation by AICAR and insulin.

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

## Additional information

### Data availability statement

The data that support the findings of the present study are available from the corresponding author upon reasonable request.

### Competing interests

The authors declare that they have no competing interests.

### Author contributions

T.E.J. and K.W.P. conceived and initiated the study. K.W.P. and T.E.J. designed the experimental set-ups. K.W.P., T.E.J., R.M.V., N.R.A., S.G., S.A.H., F.S.P. and C.H.O. performed the experiments and acquired the data. K.W.P. and T.E.J. analysed and interpreted the data. T.E.J. and K.W.P. drafted the manuscript, and all co-authors revised it critically for intellectual content. T.E.J. is the guarantor of this work, has full access to all data in the study and takes full responsibility for the integrity of the data and the accuracy of the data analyses.

### Funding

This study was funded by Lundbeck Foundation Ascending Instigator (R313-2019-643) and Novo Nordisk Foundation (NNF19OC0056839) projects to T.E.J.

### Acknowledgements

We thank Trevor Dale, University of Cardiff, and Karyn Esser, University of Florida, for providing the AXIN flox and muscle-specific tamoxifen-inducible Cre recombinase mice, and Peter Schjerling (Institute of Sports Medicine Copenhagen, Department of Orthopedic Surgery M, Bispebjerg Hospital, Copenhagen, Denmark) for performing the genotyping.

### Keywords

AMPK, exercise, glucose transport, mTOR, skeletal muscle

### Supporting information

Additional supporting information can be found online in the Supporting Information section at the end of the HTML view of the article. Supporting information files available:

**Peer Review History**

