## [Peer Review History · The Journal of Physiology]

Muscle-Specific AXIN1 and 2 double knockout does not alter AMPK/mTORC1 signaling or glucose metabolism

Kaspar Wredstrøm Persson, Roberto Meneses-Valdés, Nicoline Resen Andersen, Frederik Siem Pedersen, Samantha Gallero, Sofie Ahrens Hesselager, Carlos Henríquez Olgúin, and Thomas E Jensen

DOI: 10.1113/JP288854

Corresponding author(s): Thomas Jensen (tejensen@nxs.ku.dk)

The following individual(s) involved in review of this submission have agreed to reveal their identity: Bert Blaauw (Referee #1); Ken D O'Halloran (Referee #3)

Review Timeline:

Submission Date:	06-Mar-2025
Editorial Decision:	28-Mar-2025
Revision Received:	28-May-2025
Accepted:	11-Jun-2025

Senior Editor: Paul Greenhaff

Reviewing Editor: Bettina Mittendorfer

Transaction Report:

Dear Dr Jensen,

Re: JP-RP-2025-288854 **"Muscle-Specific AXIN1 and 2 double knockout does not alter AMPK/mTORC1 signaling or glucose metabolism"** by Kaspar Wredström Persson, Roberto Meneses-Valdés, Nicoline Resen Andersen, Frederik Siem Pedersen, Samantha Gallero, Sofie Ahrens Hesselager, Carlos Henríquez Olguín, and Thomas E Jensen

Thank you for submitting your manuscript to The Journal of Physiology. It has been assessed by a Reviewing Editor and by 3 expert referees and we are pleased to tell you that it is acceptable for publication following satisfactory revision.

REVISION CHECKLIST:

We look forward to receiving your revised submission.

Yours sincerely,

Paul Greenhaff
Senior Editor
The Journal of Physiology

REQUIRED ITEMS

- Author photo and profile. First or joint first authors are asked to provide a short biography (no more than 100 words for one author or 150 words in total for joint first authors) and a portrait photograph. These should be uploaded and clearly labelled together in a Word document with the revised version of the manuscript. See Information for Authors for further details.

- The contact information for the person responsible for 'Research Governance' at your institution needs to be provided. This includes their name and an institutional email address. Please ensure the contact is not an author on this paper and provide an alternate contact if necessary, or confirm in the submission form that the author whose email was provided has sole responsibility for research governance. This is the person who is responsible for regulations, principles and standards of good practice in research carried out at the institution, for instance the ethical treatment of animals, the keeping of proper experimental records or the reporting of results.

- You must start the Methods section with a paragraph headed Ethical approval (https://jp.msubmit.net/cgi-bin/main.plex?form_type=display_requirements#methods).

Research must comply with The Journal's policies regarding animal experiments (<https://physoc.onlinelibrary.wiley.com/hub/animal-experiments>) and adherence to these policies must be stated in the manuscript.

Authors should confirm in their Methods section that their experiments were carried out according to the guidelines laid down by their institution's animal welfare committee, including an ethics approval reference number. The Methods section must contain a statement about access to food, water and housing, details of the anaesthetic regime: anaesthetic used, dose and route of administration, and method of killing the experimental animals.

- The Journal of Physiology funds authors of provisionally accepted papers to use the premium BioRender site to create high resolution schematic figures. Follow this link and enter your details and the manuscript number to create and download figures. Upload these as the figure files for your revised submission. If you choose not to take up this offer, we require figures to be of similar quality and resolution. If you are opting out of this service to authors, state this in the Comments section on the Detailed Information page of the submission form. The link provided should only be used for the purposes of this submission. Authors will be charged for figures created on this premium BioRender account if they are not related to this manuscript submission.

- Please upload separate high-quality figure files via the submission form.

- You must upload original, uncropped western blot/gel images (including controls) if they are not included in the manuscript. This is to confirm that no inappropriate, unethical or misleading image manipulation has occurred. These should be uploaded as 'Supporting information for review process only'. Please label/highlight the original gels so that we can clearly

see which sections/lanes have been used in the manuscript figures. For more information, see:
<https://physoc.onlinelibrary.wiley.com/hub/journal-policies#imagmanip>.

- Papers must comply with the Statistics Policy: https://jp.msubmit.net/cgi-bin/main.plex?form_type=display_requirements#statistics.

In summary:

- If n {less than or equal to} 30, all data points must be plotted in the figure in a way that reveals their range and distribution. A bar graph with data points overlaid, a box and whisker plot or a violin plot (preferably with data points included) are acceptable formats.

- If $n > 30$, then the entire raw dataset must be made available either as supporting information, or hosted on a not-for-profit repository, e.g. FigShare, with access details provided in the manuscript.

- 'n' clearly defined (e.g. x cells from y slices in z animals) in the Methods. Authors should be mindful of pseudoreplication.

- All relevant 'n' values must be clearly stated in the main text, figures and tables.

- The most appropriate summary statistic (e.g. mean or median and standard deviation) must be used. Standard Error of the Mean (SEM) alone is not permitted.

- Exact p values must be stated. Authors must not use 'greater than' or 'less than'. Exact p values must be stated to three significant figures even when 'no statistical significance' is claimed.

- Please include an Abstract Figure file, as well as the Figure Legend text within the main article file. The Abstract Figure is a piece of artwork designed to give readers an immediate understanding of the research and should summarise the main conclusions. If possible, the image should be easily 'readable' from left to right or top to bottom. It should show the physiological relevance of the manuscript so readers can assess the importance and content of its findings. Abstract Figures should not merely recapitulate other figures in the manuscript. Please try to keep the diagram as simple as possible and without superfluous information that may distract from the main conclusion(s). Abstract Figures must be provided by authors no later than the revised manuscript stage and should be uploaded as a separate file during online submission labelled as File Type 'Abstract Figure'. Please also ensure that you include the figure legend in the main article file. All Abstract Figures should be created using BioRender. Authors should use The Journal's premium BioRender account to export high-resolution images. Details on how to use and access the premium account are included as part of this email.

- The corresponding author must provide an institutional email address (not a personal address) for their author account. We encourage ALL co-authors to also provide institutional email addresses. If this cannot be provided (as corresponding author), then a stamped letter must be provided from the institution which confirms their role and employment there (please upload this with the revised submission).

EDITOR COMMENTS

Senior Editor:

Thank you for the manuscript submission to The Journal of Physiology, which has been considered by a reviewing editor and two expert reviewers. The consensus opinion is positive, but both reviewers have suggested several points that could improve the impact of the manuscript. In particular, Reviewer 1 believes the manuscript could be strengthened by inclusion of data demonstrating a 'positive' physiological effect arising from the knockdown of Axin1 and Axin2, which seems to have been relatively mild. This is a helpful comment from Reviewer 1 as it will strengthen the paper. From a housekeeping perspective, please state the method of animal termination (the guidelines to authors provides further information about current requirements for animal experimentation). Similarly, please provide specific P values as per the statistical guidelines to authors. We look forward to receiving the revised manuscript.

Reviewing Editor:

The paper was reviewed by 2 experts in the field who gave high ratings and provided constructive feedback that will help the authors further improve an already strong paper to ensure it has the appropriate impact.

REFEREE COMMENTS

Referee #1:

In this manuscript the authors study the role of Axin1/2 on stimulation-dependent activation of the AMPK/mTORC1 signaling pathway(s). Axins were thought to play a critical role and to be somewhat redundant, therefore they generated an inducible muscle specific double knockout mouse model. Overall the results are mainly negative, as in, Axin1/2 does not appear to play a role in the regulation of signaling changes after contractions. Despite this negative outcome, I feel these stories can be very useful and relevant for the community.

issues:

The knockdown of Axin1 and Axin2 is relatively mild. For example in figure 2C they find the protein levels in the soleus to be down about 50%. While this is fine if you see some kind of an effect, when this is not the case it is also possible that small amounts of protein (and 50% is really not that small) can cover most of the basic tasks inside the cell. Similar observations have been reported for mTORC1 signaling. What would really make the manuscript that much stronger, is the presence of some 'positive' data. For example, are there differences in the adaptations to exercise when comparing wildtype and knockout animals? Is AMPK phosphorylation after treadmill running altered?

As AXIN has been linked to glut4 translocation and glucose uptake, are these processes altered in knockout mice, even though AMPK/mTORC1 signaling is not? Even just looking at glycogen content before and after stimulation could give some clues.

I think adding one of these things could really make the manuscript that much more interesting for the community, and could potentially help in better understanding what the signaling changes in these pathways mean with regards to certain processes. As the experimental setting is an acute one, it really offers interesting insights into acute exercise-dependent signaling changes (which is quite different from chronic gain-and-loss-of-function approaches).

Referee #2:

This is a very well-designed study by an established group. The topic is highly significant as there is some confusion in the literature regarding the role of AXIN in contraction-induced glucose uptake and the mechanisms underlying AMPK activation. Strengths of the paper include an excellent and succinct study design, strong rationale for performing the double KO experiments, quality data and figures, and supported conclusion that AXIN1/2 are dispensable for muscle glucose uptake (AICAR, Insulin, Contraction). Weaknesses include a slightly incremental study, however, the rationale is good for why this was an important experiment to conduct. It does provide stronger support for the conclusion that AXIN1/2 are not involved. In the discussion, please consider rewriting the sentence on lines 244-249 as it was unclear. Overall, this is an excellent study that will significantly add to our understanding of the mechanisms underlying glucose uptake in skeletal muscle. Also documents that we need strong physiological studies to validate some papers that have been published in cellular metabolism journals.

Referee #3: Ethics review

Thank you for submitting your manuscript to The Journal of Physiology. Some additional important details pertaining to animal welfare are required.

1. Please consult O'Halloran (2024) <https://physoc.onlinelibrary.wiley.com/doi/10.1113/JP286666> for information on requirements for ethics and welfare reporting in The Journal.

2. You must start the Methods section with the sub-heading "Ethical approval". Please edit the existing text to provide first the details of the necessary approvals, i.e., move line 96 to 86.

3. The source of the animals must be included.

4. Line 99: Please also state that the study conformed to the principles of The Journal.

5. Line 115: Please include details of the route of administration, presumably i.p. You must include details of how you determined that an adequate depth of anaesthesia was achieved before commencing surgical procedures. Include the method of killing of the animals.

6. Line 126. Include again the route of administration, how adequacy of anaesthesia was assessed, and also how frequently depth of anaesthesia was checked given that pentobarbital is a short-acting anaesthetic. Please also note that there is established concern with the administration of pentobarbital by i.p. route, as it is an irritant. The Journal of Physiology seeks to uphold the highest standards of animal welfare and is strongly encouraging authors at this juncture to consider alternative methods and may soon mandate this change for future publications.

7. Line 137: Please include details on the method of killing.

END OF COMMENTS

Senior Editor:

Thank you for the manuscript submission to The Journal of Physiology, which has been considered by a reviewing editor and two expert reviewers. The consensus opinion is positive, but both reviewers have suggested several points that could improve the impact of the manuscript. In particular, Reviewer 1 believes the manuscript could be strengthened by inclusion of data demonstrating a 'positive' physiological effect arising from the knockdown of Axin1 and Axin2, which seems to have been relatively mild. This is a helpful comment from Reviewer 1 as it will strengthen the paper. From a housekeeping perspective, please state the method of animal termination (the guidelines to authors provides further information about current requirements for animal experimentation). Similarly, please provide specific P values as per the statistical guidelines to authors. We look forward to receiving the revised manuscript.

Reviewing Editor:

The paper was reviewed by 2 experts in the field who gave high ratings and provided constructive feedback that will help the authors further improve an already strong paper to ensure it has the appropriate impact.

REFeree COMMENTS

Referee #1:

In this manuscript the authors study the role of Axin1/2 on stimulation-dependent activation of the AMPK/mTORC1 signaling pathway(s). Axins were thought to play a critical role and to be somewhat redundant, therefore they generated an inducible muscle specific double knockout mouse model. Overall the results are mainly negative, as in, Axin1/2 does not appear to play a role in the regulation of signaling changes after contractions. Despite this negative outcome, I feel these stories can be very useful and relevant for the community.

issues:

The knockdown of Axin1 and Axin2 is relatively mild. For example in figure 2C they find the protein levels in the soleus to be down about 50%. While this is fine if you see some kind of an effect, when this is not the case it is also possible that small amounts of protein (and 50% is really not that small) can cover most of the basic tasks inside the cell. Similar observations have been reported for mTORC1 signaling. What would really make the manuscript that much stronger, is the presence of some 'positive' data. For example, are there differences in the adaptations to exercise when comparing wildtype and knockout animals? Is AMPK phosphorylation after treadmill running altered?

We thank the reviewer for this thoughtful and constructive comment.

Approximately half of nuclei from skeletal muscle are known to stem from non-parenchymal cells depending on the muscle type. We previously published that an efficient ~50% muscle-specific gene excision by PCR can result in a near complete loss of KO'd protein detection or no reduction at all in a whole-muscle lysate, depending on whether the KO'd protein is dominantly expressed in non-parenchymal cells (beta-actin) or in myofibers (LKB1) (Madsen et al, AJP endo, 2018).

Presently, the reduction in AXIN1 protein levels in soleus muscle was partial (~50%), but consistent with what we previously reported in our muscle-specific AXIN1 knockout study (Li et al., 2021) using the same tamoxifen-inducible system. In our prior study, we observed a similar ~50% decrease

in AXIN1 protein in skeletal muscle lysates, which we attributed to residual non-muscle AXIN1 cell content. In other muscles, the reduction in AXIN1 protein in the AXIN1 KO vs. WT was much larger, reaching 80-90%, similar to what we saw in the present study. Based on these observations, we believe that the partial reduction is not due to ineffective excision but rather AXIN1 expression in non-parenchymal cells.

While we agree that AXIN proteins might have exerted additional effects under other physiological or stress conditions, our current study was designed to address a focused question arising from the possibility of AXIN2 compensation in the absence of AXIN1 (Li et al., 2021): Is AXIN1/2 required for acute AMPK and mTORC1 signaling, or for AICAR- and insulin-stimulated glucose uptake in skeletal muscle? The absence of genotype effects across these conditions strongly supports the conclusion that AXIN1/2 are dispensable for these acute responses. The absence of any meaningful genotype difference also provide little ethical or economical justification to pursue further characterization and we stopped breeding the AXIN1/2 double KO mice.

As AXIN has been linked to glut4 translocation and glucose uptaked, are these processes altered in knockout mice, even though AMPK/mTORC1 signaling is not? Even just looking at glycogen content before and after stimulation could give some clues.

We agree that this is a very interesting line of inquiry. We would have enthusiastically pursued analyses of GLUT4 translocation or glycogen metabolism if our glucose uptake data had revealed any genotype effect. However, since both AICAR- and insulin-stimulated glucose uptake were entirely unaffected by AXIN1/2 deletion, we found little rationale to investigate downstream processes such as GLUT4 translocation or glycogen content.

While AXIN1 has been implicated in GLUT4 trafficking in cultured adipocytes and muscle cells (Guo et al., 2012), our results suggest that adult skeletal muscle glucose uptake is intact in the absence of both AXIN1 and AXIN2, at least in the acute setting.

I think adding one of these things could really make the manuscript that much more interesting for the community, and could potentially help in better understanding what the signaling changes in these pathways mean with regards to certain processes. As the experimental setting is an acute one, it really offers interesting insights into acute exercise-depepndent signaling changes (which is quite different from chronic gain-and-loss-of-function approaches.

We agree in principle that more could have been done and that chronic or adaptive models might have revealed some roles for AXIN proteins. However, our aim was to test the AXIN protein requirement for acute AMPK/mTORC1 signalling and glucose uptake. We hope the present findings provide a useful foundation for future studies exploring long-term adaptations.

Referee #2:

This is a very well-designed study by an established group. The topic is highly significant as there is some confusion in the literature regarding the role of AXIN in contraction-induced glucose uptake and the mechanisms underlying AMPK activation. Strengths of the paper include an excellent and succinct study design, strong rationale for performing the double KO experiments, quality data and figures, and supported conclusion that AXIN1/2 are dispensable for muscle glucose uptake (AICAR, Insulin, Contraction). Weaknesses include a slightly incremental study, however, the rationale is good for why this was an important experiment to conduct. It does provide stronger support for the conclusion that AXIN1/2 are not involved. In the discussion, please consider rewriting the sentence

on lines 244-249 as it was unclear. Overall, this is an excellent study that will significantly add to our understanding of the mechanisms underlying glucose uptake in skeletal muscle. Also documents that we need strong physiological studies to validate some papers that have been published in cellular metabolism journals.

We sincerely thank the reviewer for the thoughtful and encouraging comments on our study. We appreciate the suggestion to clarify the sentence on lines 244–249, and we agree that the original wording could be improved. We have now revised this section of the discussion to more clearly link our findings to the conclusion and better articulate why AXIN-dependent AMPK regulation is unlikely to play a major role in contraction-induced AMPK activation or mTORC1 suppression in skeletal muscle. The revised sentence now reads:

“Together, these findings argue against a significant role for AXIN-dependent AMPK regulation in contraction-induced AMPK activation or the associated suppression of mTORC1 signaling in skeletal muscle.”

Referee #3: Ethics review

Thank you for submitting your manuscript to The Journal of Physiology. Some additional important details pertaining to animal welfare are required.

1. Please consult O'Halloran (2024) <https://physoc.onlinelibrary.wiley.com/doi/10.1113/JP286666> for information on requirements for ethics and welfare reporting in The Journal.

2. You must start the Methods section with the sub-heading "Ethical approval". Please edit the existing text to provide first the details of the necessary approvals, i.e., move line 96 to 86.

3. The source of the animals must be included.

4. Line 99: Please also state that the study conformed to the principles of The Journal.

5. Line 115: Please include details of the route of administration, presumably i.p. You must include details of how you determined that an adequate depth of anaesthesia was achieved before commencing surgical procedures. Include the method of killing of the animals

6. Line 126. Include again the route of administration, how adequacy of anaesthesia was assessed, and also how frequently depth of anaesthesia was checked given that pentobarbital is a short-acting anaesthetic. Please also note that there is established concern with the administration of pentobarbital by i.p. route, as it is an irritant. The Journal of Physiology seeks to uphold the highest standards of animal welfare and is strongly encouraging authors at this juncture to consider alternative methods and may soon mandate this change for future publications.

7. Line 137: Please include details on the method of killing.

We thank the reviewer for the thorough evaluation and for highlighting the Journal's standards regarding ethics and animal welfare reporting. We have carefully revised the manuscript to address all points:

1. **O'Halloran (2024)** was consulted and used as a guide for the revisions.

2. A new subheading “**Ethical approval**” has been added at the start of the *Methods* section, and the relevant approval details have been moved accordingly.
3. The **source of the animals** have been included.
4. A statement confirming that the study conformed to the principles of *The Journal of Physiology* has been included.
5. For all procedures, the **route and dose of anaesthesia** are now clearly stated, along with the method of confirming **adequate anaesthetic depth** and the method of **humane killing**.
6. These details are also repeated for **in situ contraction experiments**, with additional clarification that cervical dislocation was performed under deep anaesthesia. We acknowledge the Journal’s guidance on the use of pentobarbital via the i.p. route and will consider alternative approaches in future studies.
7. The **method of killing** has been explicitly stated for all experimental cohorts.

We hope these revisions satisfactorily address the reviewer’s concerns.

Dear Professor Jensen,

Re: JP-RP-2025-288854R1 "**Muscle-Specific AXIN1 and 2 double knockout does not alter AMPK/mTORC1 signaling or glucose metabolism**" by Kaspar Wredström Persson, Roberto Meneses-Valdés, Nicoline Resen Andersen, Frederik Siem Pedersen, Samantha Gallero, Sofie Ahrens Hesselager, Carlos Henríquez Olguín, and Thomas E Jensen

We are pleased to tell you that your paper has been accepted for publication in The Journal of Physiology.

Yours sincerely,

Paul Greenhaff
Senior Editor
The Journal of Physiology

If you would like to receive our 'Research Roundup', a monthly newsletter highlighting the cutting-edge research published in The Physiological Society's family of journals (The Journal of Physiology, Experimental Physiology, Physiological Reports, The Journal of Nutritional Physiology and The Journal of Precision Medicine: Health and Disease), please click this link, fill in your name and email address and select 'Research Roundup':
<https://www.physoc.org/journals-and-media/membernews>

- You can help your research get the attention it deserves! Check out Wiley's free Promotion Guide for best-practice recommendations for promoting your work at: www.wileyauthors.com/eeo/guide. You can learn more about Wiley Editing Services which offers professional video, design, and writing services to create shareable video abstracts, infographics, conference posters, lay summaries, and research news stories for your research at: www.wileyauthors.com/eeo/promotion.

EDITOR COMMENTS

Senior Editor:

Thank you for the revised manuscript and author response which have both been considered by the same reviewing editor and reviewers that considered the original submission. All are of the opinion that the authors have addressed the concerns previously raised and that the revised manuscript is deemed to be acceptable for publication. Congratulations and thank you

for considering The Journal of Physiology to your publish your research.

REFEREE COMMENTS

Referee #1:

The authors responded to all the points raised in an acceptable manner

Referee #2:

The authors have addressed my original concerns.